# Accuracy of Inflow Inversion Recovery (IFIR) for Upper Abdominal Arteries Evaluation: Comparison with Contrast-Enhanced MR and CTA

**DOI:** 10.3390/diagnostics12040825

**Published:** 2022-03-28

**Authors:** Roberto Simonini, Pietro Andrea Bonaffini, Marco Porta, Cesare Maino, Francesco Saverio Carbone, Ludovico Dulcetta, Paolo Brambilla, Paolo Marra, Sandro Sironi

**Affiliations:** 1Department of Radiology, ASST Papa Giovanni XXIII Hospital, Piazza OMS 1, 24127 Bergamo, Italy; roberto.simonini89@gmail.com (R.S.); pa.bonaffini@gmail.com (P.A.B.); f.carbone15@campus.unimib.it (F.S.C.); l.dulcetta@campus.unimib.it (L.D.); pbrambilla@asst-pg23.it (P.B.); pmarra@asst-pg23.it (P.M.); sandro.sironi@unimib.it (S.S.); 2School of Medicine, University Milano Bicocca, Piazza dell’Ateneo Nuovo, 1, 20126 Milano, Italy; 3Department of Radiology, San Gerardo Hospital, Via G. B. Pergolesi 33, 20900 Monza, Italy; mainocesare@gmail.com

**Keywords:** non-contrast magnetic resonance angiography, magnetic resonance angiography, inflow-sensitive inversion recovery technique, computed tomography angiography, hepatic and visceral artery, liver transplantation

## Abstract

**Background:** Inflow-sensitive inversion recovery (IFIR) is a recently introduced technique to perform unenhanced magnetic resonance angiography (MRA). The purpose of our study is to determine the accuracy of IFIR-MRA in the evaluation of upper abdominal arteries, compared to standard MRA and computed tomography angiography (CTA). **Materials and Methods:** Seventy patients undergoing upper abdomen Magnetic Resonance Imaging (MRI) in different clinical settings were enrolled. The MRI protocol included an IFIR-MRA sequence that was intra-individually compared by using a qualitative 4-point scale in the same patients who underwent concomitant or close MRA (*n* = 65) and/or CTA (*n* = 44). Celiac trunk (CA), common-proper-left-right hepatic artery (C-P-L-R-HA), left gastric artery (LGA), gastroduodenal artery (GDA), splenic artery (SA), renal arteries (RA) and superior mesenteric artery (SMA) were assessed. **Results:** IFIR-MRA images were better rated in comparison with MRA. Particularly, all arteries obtained a statistically significant higher qualitative rating value (all *p* < 0.05). IFIR-MRA and MRA exhibited acceptable intraclass correlation coefficients (ICC) values for CA, C-L-R-HA, and SMA (ICC 0.507, 0.591, 0.615, 0.570, 0.525). IFIR-MRA and CTA showed significant correlations in C-P-L-R-HA (τ = 0.362, 0.261, 0.308, 0.307, respectively; *p* < 0.05), and in RA (τ = 0.279, *p* < 0.05). **Conclusions:** Compared to MRA, IFIR-MRA demonstrated a higher image quality in the majority of upper abdomen arterial vessels assessment. LHA and RHA branches could be better visualized with IFIR sequences, when visualizable. Based on these findings, we suggest to routinely integrate IFIR sequences in upper abdomen MRI studies.

## 1. Introduction

Evaluation of visceral artery anatomy and patency is essential in several conditions affecting the upper abdomen such as liver resection in hepatobiliary surgery, liver transplantation (LT), or pancreatic head carcinoma staging [1,2].

Selective catheter digital subtraction angiography (DSA) and contrast-enhanced computed tomography angiography (CTA) are considered the reference standard methods for evaluating visceral arteries anatomy and their variants [3,4,5]. However, both techniques present some disadvantages. DSA is burdened with a greater dose of radiation and contrast media, higher cost, and invasiveness in comparison with CTA, and cannot be used for screening or routine follow-up [3]. Then, both use ionizing radiations, and require iodinated contrast agents, the latter being particularly limiting in patients with compromised renal function.

Thus, magnetic resonance angiography (MRA) could represent a valid alternative to these techniques, considering also that the gadolinium-based contrast agents are generally safer than the iodinated ones used for contrast-enhanced computed tomography (CT) with no nephrotoxicity and with extremely rare anaphylactic reactions. However, nephrogenic systemic fibrosis has been reported as complication in patients with renal failure who received intravenous gadolinium injection [6], although new macrocyclic contrast agents provide higher chelate stability compared to linear gadolinium-based contrast agents with substantially minimum or none release of Gd ions, known to be responsible for nephrogenic systemic fibrosis (NSF) [7]. 

The unenhanced respiratory-gated magnetization-prepared three-dimensional (3D) steady-state-free-precession (SSFP) inflow-sensitive inversion recovery (IFIR) technique in MRA has recently showed reproducible and promising results in the evaluation of upper abdomen arteries. It uses a preparatory inversion pulse to reduce signals from static tissue, while leaving inflow arterial blood unaffected, resulting in sparse arterial vasculature on modest tissue background [8,9]. 

Furthermore, IFIR-MRA do not require breath holding, which represents a problem in gradient-echo based magnetic resonance (MR) sequences, known to be susceptible to motion artifacts. In fact, the ability to perform a 20 s breath hold is one of the key requirements for abdominal contrast enhanced MRA to avoid breathing artifacts, thus leading in deterioration of image quality in patients with compromised pulmonary function or suffering liver cirrhosis and ascites.

Another artifact which can affect MRA sequences using gadolinium-based contrast media is Gibbs phenomenon, also known as “truncation artifact”, which manifests as ringing false images in the final study, affecting the visualization of finest arteries [10].

The aim of our study is to determine the accuracy of contrast enhanced MRA and unenhanced IFIR-MRA in the evaluation of upper abdomen arteries, and to compare both to CTA, considered as the reference standard.

## 2. Materials and Methods

### 2.1. Population of Study

From November 2019 to April 2020, we retrospectively enrolled consecutive patients undergoing a magnetic resonance imaging (MRI) study of the upper abdomen, either with or without contrast. All patients gave written consent for MRI and their anonymity was granted. We enrolled a total of 70 patients: twelve patients underwent upper abdominal MRI for extra-hepatic indications, 8/12 characterization of focal pancreatic lesions, 1/12 evaluation of abdominal aorta, 2/12 follow up in non-hepatic neoplastic pathology, 1/12 characterization of an adrenal mass. Fifty-eight patients underwent MRI for hepatic indications, 29/58 characterization of focal hepatic lesions, 13/58 follow up in hepatocellular carcinoma, 6/58 autoimmune hepatic disease, 6/58 biliary disease, 4/58 follow up in LT. All MRI studies included IFIR sequence; in addition, 39/70 patients underwent MRA and CTA in a close timing, 26/70 patients performed only MRA and 5/70 only CTA (Figure 1).

Exclusion criteria of the study were: age < 18 years, renal insufficiency (indicated by an estimated glomerular filtration rate of 30 mL/min/1.73 m^2^), cardiac pacemaker and other implants and devices not approved for MR, claustrophobia, pregnancy, and studies with degraded image quality.

### 2.2. Imaging Technique

#### 2.2.1. IFIR-MRA

MRI was performed on a 1.5 Tesla MR scanner (Optima MR450w, GE Healthcare, Little Chalfont, UK) equipped with an 8-channel Body Array Coils (MEDRAD 8-Channel eCoil, GE Healthcare, Little Chalfont, UK). All patients were examined in supine position, with both arms raised above their heads and using respiratory triggering; no cardiac gating was needed. The MRI protocol included standard sequences for liver or pancreas assessment, with dedicated cholangiography sequences in selected cases (bile ducts, main pancreatic duct, pancreatic cysts). 

IFIR sequence was planned in the axial plane, with no cardiac or pulse gating and centered at the liver hilum, covering the whole hepatic region. The sequence used a spatially selective inversion pulse covering the heart, descending aorta, hepatic veins, and portal veins to suppress the other inflows and static tissue signals and acquired data using a 3D balanced steady-state-free-precession (SSFP) acquisition with chemical fat suppression. Scanning parameters were the following: TR 4.75 ms; TE 2.37 ms; flip angle 90°, TI 214 ms, matrix 256 × 256 (512 × 512); field of view (FOV) 36 × 40 cm according to body size; slice thickness 2.0 mm (1 mm); slice number 60 interpolated; NEX 1.

#### 2.2.2. MRA

MRA study included the whole liver region, and the sequences were performed in an axial plane. The scanning parameters of MRA were TR 1⁄4 3.7 ms, TE 1⁄4 1.2 ms, flip angle (FA) 1⁄4 30, TI 1⁄4 14 ms, matrix 288 × 192, FOV 36 × 36 cm, slice thickness 3 mm. Intravenous administration of gadolinium-based contrast media was performed manually and the scan for MRA was started visually when contrast media reached the abdominal aorta.

#### 2.2.3. CTA

CTA was performed on a 64-slice scanner (Philips Brilliance; Philips Medical Systems, Best, The Netherlands) with the following scan parameters: 120 kV, automated tube-current modulation (based on scout image), collimation 64 × 0.625 mm; rotation time 0.5 s; thickness 2 mm; increment 1 mm. FOV 350 mm, matrix 512 × 512. The start of scanning was individually obtained for each patient during intravenous iodinated contrast media administration (Iomeprol, 350 mgI/mL; flow rate 4 mL/s, volume based on body weight) by using bolus–tracking technique, with a trigger level of 150 HU placed at the supra-renal aorta and a delay time of 15 s for arterial phase liver study.

### 2.3. Imaging Analysis

Comparison with MRA (65 patients) and CTA (44 patients) was independently performed by two radiologists, with 3 years of experience in abdominal imaging. They qualitatively evaluated images by using a 4-point Likert scale [11]: score 4 (excellent image quality without any image quality-deteriorating artifacts such as motion or blurring), 3 (good image quality with presence of image quality-deteriorating artifacts but with possible evaluation and good confidence), 2 (adequate image quality with the presence of image quality-deteriorating artifacts but with possible evaluation and moderate confidence) and 1 (poor image quality with no evaluation possible). The following main arterial branches were assessed: celiac trunk (CA), common-proper-left-right hepatic artery (C-P-L-R-HA), left gastric artery (LGA), gastroduodenal artery (GDA), splenic artery (SA), renal arteries (RA), and superior mesenteric artery (SMA). IFIR images were blindly analyzed first, then MRA and CTA for confirmation. Windowing, multiplanar reformations, and volume rendering were additionally employed upon radiologist discretion. The two readers looked for the presence of variants of visceral arteries, especially hepatic branches classified according to the Michels’ classification system as follows: Type I normal anatomy, Type II replaced LHA from the LGA, Type III replaced RHA from the SMA, Type IV replaced RHA and LHA, type V accessory LHA, Type VI accessory RHA, Type VII accessory RHA and LHA Type VIII replaced RHA or LHA with other hepatic artery being an accessory one, Type IX: hepatic trunk as a branch of the SMA, Type X CHA from the LGA. Variants not included in ten types by Michels’ were classified as Type XI [12].

### 2.4. Statistical Analysis

Categorical variables were presented using median and interquartile range (IQR) values, while continuous variables as mean ± standard deviation (SD). Categorical variables between groups (IFIR-MRA, MRA, and CTA) were compared by using the Mann-Whitney U test or Kruskal-Wallis test, as appropriate. Intraclass correlation coefficients (ICC) and their 95% confidence interval (CI) were calculated based on a mean-rating (k = 2), absolute-agreement, 2-way mixed-effects model. Moreover, Kendall’s Tau (τ) test was performed between IFIR-MRA, MRA, and CTA values, setting the last as the reference standard. The τ correlation coefficient returns a value of 0 to 1, where 0 represents no relationship, 1 a perfect positive relationship. 95%CI was calculated using bootstrap with 500 iterations and random number seed 978. For both inter-test qualitative reliability and inter-reader agreement, ICC values less than 0.5, between 0.5 and 0.75, between 0.75 and 0.9, and greater than 0.90 are indicative of poor, moderate, good, and excellent reliability, respectively. All tests were two-sided, and the *p*-value ≤ 0.05 was considered statistically significant. All the statistical analyses were performed by using IBM SPSS 26.0 (SPSS Incorporated, Chicago, IL, USA) and MedCalc Statistical Software 19.5 (MedCalc Software, Ostend, Belgium).

## 3. Results

### 3.1. Patient Population

By applying inclusion end exclusion criteria, we enrolled a total of 70 patients. The majority was male (M/F = 45/25), with a mean age of 58 years (SD ± 14, range 12–86). 

All patients underwent IFIR-MRA (*n* = 70). In most cases (*n* = 65) paramagnetic contrast medium was injected (*n* = 28 Gd-BOPTA, *n* = 32 Gd-EOB-DTPA, *n* = 5 Gadoteridol). A total of 44 patients also underwent a CTA with a median time interval between examinations of 8 months (IQR 2–12), either before or after MRI (Figure 1).

### 3.2. Inter-Reader Agreement

Overall, the agreement between the 2 readers was very good regarding all imaging techniques. IFIR-MRA, compared with MRA, showed a good ICC for all arteries except for PHA, LGA, and SA (ICCs 0.834 vs. 0.836, 0.722 vs. 0.770, 0.700 vs. 0.705, respectively). In comparison with CTA, IFIR-MRA showed a higher ICC only for CA (0.801 vs. 0.795).

### 3.3. Overall Qualitative Analysis

At IFIR-MRA, CHA, GDA, LGA, and SA were correctly evaluated in all patients, while the CA, PHA, and RHA in the majority (*n* = 69, 98.5%). RA were correctly evaluated in 57 patients (81.4%). Regarding image quality, CA, CHA, SA, and RA obtained a median value of 4 (IQR 2–4), GDA and PHA of 3 (IQR 2–4), while LHA, RHA, and LGA of 2 (IQR 1–4) (Table 1).

At MRA, both in Gd-BOPTA and Gd-EOB-DTPA images, all vessels were correctly evaluated in all patients. CA, CHA, and SA obtained a median value of 3 (IQR 2–4), while GDA, PHA, LGA, and RA have a median value of 2 (IQR 1–3). Interestingly, LHA and RHA obtained the lowest median value 1 (IQR 1–2).

At CTA, all vessels were correctly evaluated in all patients, showing a median value of 4 with all IQRs of 3–4.

### 3.4. Pairwise Comparisons

Overall IFIR-MRA images were better rated in comparison with MRA ones. All arteries obtained a statistically significant higher qualitative rating value (all *p* < 0.05). The comparison between Gd-BOPTA and Gd-EOB-DTPA MRA did not show any significant difference regarding quality between all arteries analyzed (all *p* > 0.05).

The comparison between IFIR-MRA and MRA reported an overall good inter-test reliability, with acceptable ICC values for the CA, CHA, LHA, and RHA, and SMA (ICC values of 0.507, 0.591, 0.615, 0.570, 0.525), while the agreement was poor for the remaining branches (all ICC < 0.5) (Table 2). However, our analysis showed significant correlations in all examined arteries, exception of RA. In particular, the CHA, LHA, and RHA showed the highest ICC between the two MRI techniques (τ = 0.362, τ = 0.416, τ = 0.393, respectively, all *p* < 0.05).

Even if the inter-test reliability between IFIR-MRA and CTA was good only for the CHA (ICC = 0.552), on the other hand the comparison between the two techniques reported significant correlations in the evaluation of more arteries, such as C-P-L-R-HA (τ = 0.362, τ = 0.261, τ = 0.308, τ = 0.307, respectively, all *p* < 0.05), and in RA (τ = 0.279, *p* < 0.05).

Conversely, MRA did not show comparable correlations with CTA, both in terms of inter-test reliability and correlation, except the last for PHA, LGA, and RA (τ = 0.320, τ = 0.220, τ = 0.216, respectively, all *p* < 0.05). All correlations with 95%CI are reported in Table 3.

### 3.5. Anatomic Vessels Variants

Visceral abdominal branches variants are shown in Table 4. We also observed other anatomical variants concerning the RA, in particular: 2 left RA (6 patients); 2 right RA (9 patients); 3 right RA (1 patient). In 2 patients we found simultaneously double right, and double left RA.

## 4. Discussion

Arterial anatomy and patency assessment is fundamental for pre-operatory planning or follow-up in upper abdomen surgery, such as in LT patients. CTA or MRA are usually the preferred modalities [6]. However, the US Food and Drug Administration and UK Commission on Human Medicines recommended banning gadolinium use in patients with renal failure, in those who underwent LT or are on the transplant waiting, because of the possibility of hepatorenal syndrome. On the other hand, CTA offers a fast and reliable method for high quality vessels anatomy evaluation. However, it is associated with similar concerns about iodine contrast means use in patients with poor renal function. Moreover, the reduction of ionizing radiations exposure should be a main target, especially in pediatric patients. All these factors make non-contrast-enhanced MRA a good choice to replace conventional MRA with gadolinium and CTA, avoiding all contrast-related risks, problems with breath holding and radiation dose exposure.

IFIR-MRA has been proven to give optimal visualization of hepatic portal veins [13,14] and HA [9], RA [15] and supra-aortic arteries [16].

Our results shows that IFIR-MRA is superior to MRA in terms of quality for visualization of upper abdomen arteries, in particular the main left and right hepatic branches, when visualizable. These results are in line with other previous studies that particularly evaluated the RA and other main upper abdomen arteries, such as CA, SMA, SA, CHA [17,18], but in contrast with other papers, where MRA seemed to be superior to IFIR-MRA, especially in the assessment of visceral and HA [9,19]. These differences among Authors could be partially explained by the different centering of the slab volume, which may differ between the origin of the CA and the RA [18,19,20,21]. In fact, it should be centered just above the origin of the vessel of interest to minimize the progressive saturation of the magnetization of inflowing arterial spins once they have entered the imaging volume. Another explanation to the difference in image quality among MRA studies can be explained by the difference in gadolinium flow-velocity injection, which may generate ring artifacts affecting the image quality (10), varying between 2 and 3 mL/s [19,20] or manually, as in our case. Then, a signal loss towards the periphery of these vessels can be observed due to the length and tortuosity of SA, HA, and GDA, which differ from the short course of the RA. Finally, turbulent blood flow may lead to intravoxel dephasing within the vessel, resembling another potential cause of signal loss within the artery.

In a previous study, the accuracy of IFIR-MRA and MRA was determined by comparing these techniques with DSA [19], but the latter is not routinely used due to its invasiveness and costs. Therefore, to the best of our knowledge, this is the first study comparing the accuracy of all commonest and non-invasive techniques to assess upper abdomen arteries: IFIR-MRA, MRA, and standard CTA. CTA has revealed, as expected, to be superior to MRA in the depiction of upper abdominal arteries. However, IFIR-MRA and CTA reported a significant correlation for C-P-L-R-HA and RA, while MRA did not show comparable correlations with CTA, suggesting that MRA provided the lowest diagnostic image quality. The most important clinical impact of our results is the possibility to assemble an integrated imaging workup in liver transplant patients, which requires a life-long follow-up. These patients might benefit from focused ultrasound assessment of main vessels waveforms and a dedicated MR protocol without contrast injection, preserving renal function and ionizing radiation exposure. Specifically, cholangiography sequences and IFIR might allow a proper assessment of bile ducts and main arterial vessels, respectively, in a single session.

In our population, we observed several visceral vessels variants. Type I turned out to be the most common pattern, being present in 47 patients (67%), according to the incidence reported in literature (55% to 81%) [12,22]. The other variants were distributed as follows: Type V in 10 patients (14%); Type VI in 6 patients (9%); Type III in 2 patients (3%); Type IX in 2 patients (3%); Type XI in 2 patients (3%) (Figure 2); Type VII in 1 patient (1%).

The major limit of this retrospective study is that CTA was not performed in all cases (44/70) and basically for arterial phase liver assessment, and not directly for angiographic purposes; however, we achieved an optimal visualization of all arterial vessels, enabling us to consider CTA as gold standard imaging. Secondly, median time interval between CTA and IFIR-MRA was 8 months, but no significant arterial pathology (as potential bias) has been depicted in our population. Lastly, the heterogeneous study population might have introduced some selection bias.

In conclusion, this study showed interesting results for the potential role of IFIR-MRA as a reproducible, non-invasive and radiation free technique to assess upper abdominal arterial vessels in the routine clinical practice. IFIR-MRA may be also useful if the contrast medium is contraindicated. Avoiding the need for contrast material has several advantages, including overall healthcare cost savings, reducing delays due to peripheral intravenous cannula placement, and removing any potential risks related to toxicity and allergic reactions. Compared to MRA, IFIR-MRA demonstrated a higher image quality in main upper abdomen arterial vessels assessment. In particular, LHA and RHA branches could be better visualized with IFIR sequences. This might suggest to routinely integrate this sequence in upper abdomen MR studies along with cholangiography sequences, allowing radiologists even to avoid contrast injection, especially in liver transplant patients. Furthermore, pediatric population can represent a potential area of application after sequence optimization in future studies.

## Figures and Tables

**Figure 1 diagnostics-12-00825-f001:**
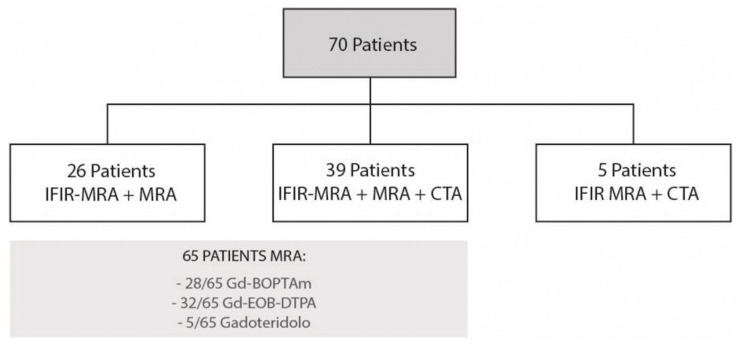
Flow diagram for patient selection.

**Figure 2 diagnostics-12-00825-f002:**
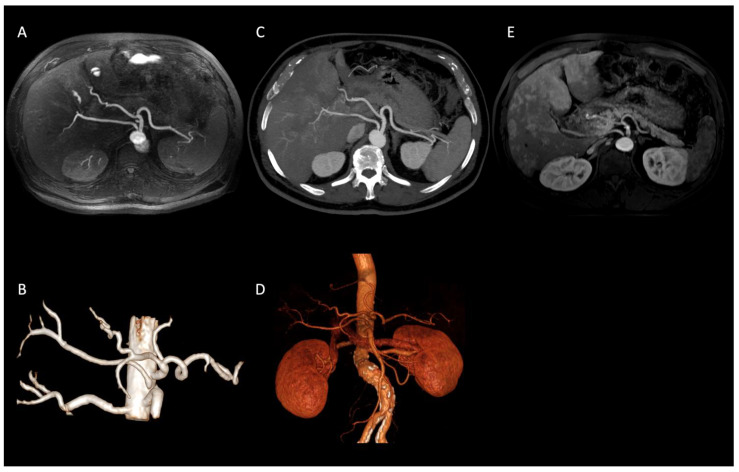
Axial IFIR-MRA (**A**) and CTA (**B**) MIP reconstructions properly showing the direct origin of the right hepatic artery from the aorta (Type XI according to Michels’ classification system) and the remaining main celiac trunk branches up to the proximal bifurcations (spleen and hepatic hilum). On MRA (**C**) MIP reconstruction image quality and resolution are lower, not allowing the proper assessment of proximal bifurcations. IFIR-MRA (**D**) and CTA (**E**) 3D reconstructions, showing a comparable diagnostic performance.

**Table 1 diagnostics-12-00825-t001:** Median and IQR values of subjective image quality of IFIR-MRA, MRA and CTA images by the most experienced reader. IFIR-MRA images obtained an overall higher image quality in comparison with MRA in all analyzed arteries.

	IFIR-MRA	MRA	*p*-Value	CTA
**Celiac trunk (median, IQR)**	4 (3–4)	3 (3–4)	<0.0001	4 (4)
**Common hepatic artery (median, IQR)**	4 (2–4)	3 (2–3)	<0.0001	4 (3–4)
**Gastroduodenal artery (median, IQR)**	3 (2–4)	2 (1–2)	<0.0001	4 (3–4)
**Proper hepatic artery (median, IQR)**	3 (2–4)	2 (2–3)	<0.0001	4 (3–4)
**Left hepatic artery (median, IQR)**	2 (1–3)	1 (1–2)	0.003	4 (3–4)
**Right hepatic artery (median, IQR)**	2 (1–4)	1 (1–2)	0.001	4 (3–4)
**Left gastric artery (median, IQR)**	2 (2–3)	2 (1–2)	0.045	4 (3–4)
**Splenic artery (median, IQR)**	4 (3–4)	3 (2–3)	<0.0001	4 (4)
**Superior mesenteric artery (median, IQR)**	4 (3–4)	2 (2–3)	<0.0001	4 (4)
**Renal arteries (median, IQR)**	4 (3–4)	2 (2–3)	<0.0001	4 (4)

**Table 2 diagnostics-12-00825-t002:** Intraclass correlation coefficients between the two readers regarding imaging quality of IFIR-MR, MRA and CTA.

	Intraclass Correlation Coefficient (95%CI)		
	IFIR-MRA	MRA	CTA
**Celiac trunk**	0.801 (0.696–0.871)	0.630 (0.393–0.774)	0.795 (0.606–0.893)
**Common hepatic artery**	0.873 (0.802–0.919)	0.714 (0.531–0.825)	0.889 (0.814–0.922)
**Gastroduodenal artery**	0.798 (0.693–0.869)	0.774 (0.629–0.862)	0.877 (0.813–0.919)
**Proper hepatic artery**	0.834 (0.744–0.894)	0.846 (0.749–0.906)	0.891 (0.801–0.944)
**Left hepatic artery**	0.823 (0.719–0.891)	0.749 (0.582- 0.849)	0.911 (0.901–0.933)
**Right hepatic artery**	0.779 (0.664–0.858)	0.719 (0.538- 0.830)	0.899 (0.867–0.954)
**Left gastric artery**	0.722 (0.588–0.818)	0.770 (0.622–0.860)	0.857 (0.831–0.900)
**Splenic artery**	0.700 (0.558–0.802)	0.705 (0.516–0.820)	0.755 (0.745–0.802)
**Superior mesenteric artery**	0.701 (0.555- 0.806)	0.437 (0.078–0.657)	0.742 (0.713–0.788)
**Renal arteries**	0.784 (0.656–0.868)	0.674 (0.462–0.803)	0.798 (0.766–0.845)

**Table 3 diagnostics-12-00825-t003:** ICC values based on absolute-agreement, 2-way mixed-effects model, and Tau correlation coefficients between IFIR-MRA, MRA and CTA.

	IFIR-MRA vs. MRA			IFIR-MRA vs. CTA			MRA vs. CTA		
	ICC (95%CI)	τ (95%CI)	*p*-Value	ICC (95%CI)	τ (95%CI)	*p*-Value	ICC (95%CI)	τ (95%CI)	*p*-Value
**Celiac trunk**	0.507 (0.189–0.700)	0.341 (0.123–0.519)	<0.0001	0.076 (−0.719–0.425)	0.035 (0.222–1.000)	0.710	0.240 (−0.424–0.594)	0.205 (0.048–1.000)	0.064
**Common hepatic artery**	0.591 (0.329–0.750)	0.362 (0.142–0.564)	0.002	0.552 (0.167–0.759)	0.362 (0.078–0.555)	0.008	0.213 (−0.475–0.580)	0.101 (−0.170–0.437)	0.363
**Gastro-duodenal artery**	0.464 (0.121–0.673)	0.311 (0.116–0.504)	0.003	0.261 (−0.384–0.606)	0.115 (−0.174–0.379)	0.297	0.120 (−0.663–0.534)	0.073 (−0.297–0.360)	0.520
**Proper hepatic artery**	0.360 (0.048–0.610)	0.207 (0.048–0.408)	0.014	0.282 (−0.345–0.617)	0.261 (−0.021–0.511)	0.017	0.470 (−0.133–0.719)	0.320 (−0.026–0.533)	0.003
**Left hepatic artery**	0.615 (0.363–0.767)	0.416 (0.188–0.584)	<0.0001	0.387 (−0.168–0.678)	0.308 (−0.059–0.508)	0.006	0.249 (−0.443–0.610)	0.174 (−0.162–0.419)	0.192
**Right hepatic artery**	0.570 (0.296–0.738)	0.393 (0.155–0.569)	<0.0001	0.443 (−0.043–0.703)	0.370 (0.073–0.550)	0.001	0.158 (−0.590–0.555)	0.103 (−0.214–0.382)	0.359
**Left gastric artery**	0.310 (0.130–0.579)	0.128 (−0.146–0.366)	0.033	0.208 (−0.485–0.577)	0.093 (−0.175–0.336)	0.403	0.382 (−0.166–0.673)	0.220 (−0.036–0.471)	0.047
**Splenic artery**	0.462 (0.117–0.671)	0.243 (0.020–0.441)	0.042	0.423 (−0.072–0.690)	0.177 (−0.188–0.506)	0.104	0.125 (−0.639–0.533)	0.067 (−0.344–0.448)	0.556
**Superior mesenteric artery**	0.525 (0.219–0.711)	0.240 (0.023–0.498)	0.005	−0.122 (−0.071–0.690)	0.130 (−0.230–1.00)	0.314	0.012 (−0.914–0.464)	0.054 (−0.103–0.633)	0.320
**Renal arteries**	0.257 (−0.294–0.573)	0.138 (0.079–0.324)	0.152	0.315 (−0.401–0.666)	0.279 (−0.113–0.645)	0.027	0.364 (−0.222–0.669)	0.216 (−0.117–0.462)	0.041

**Table 4 diagnostics-12-00825-t004:** Percentage of hepatic artery variants according to Michels’ classification system.

Michels’ Classification System: Hepatic Arteries Variants	
Type I (normal pattern)	67%
Type V (an accessory LHA)	14%
Type VI (an accessory RHA)	9%
Type III (a replaced RHA from the SMA)	3%
Type IX (the hepatic trunk as a branch of the SMA)	3%
Type XI (not included in other types)	3%
Type VII (accessory RHA and LHA)	1%

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
