# Peer review of "Accuracy of Inflow Inversion Recovery (IFIR) for Upper Abdominal Arteries Evaluation: Comparison with Contrast-Enhanced MR and CTA"

_diagnostics, 2022, doi:10.3390/diagnostics12040825_

Round 1

Reviewer 1 Report

The manuscript compares the accuracy of Inflow Inversion Recovery (IFIR) sequences, a new unenhanced MR-angiography, with CTA and traditional gadolinium enhanced MRA. The manuscript is well written and presented.

I can suggest only minor changes:

1) in the method section, to be consistent, when mentioning contrast agents please use the name of the molecule instead that commercial name ("Iomeron (Guerbet??..please check)". Alternatively you should use commercial names for gadolinium based agents as well.

2) in the methods please mention the acquisition plane for IFIR and MRA sequences and whether any cardiac or pulse gating was necessary.

3) in the discussion section, in order to give a comprehensive overview of this application, please mention the possibilities of IFIR sequences in other anatomic districts with appropriate references.

Author Response

Point 1: in the method section, to be consistent, when mentioning contrast agents please use the name of the molecule instead that commercial name ("Iomeron (Guerbet??..please check)". Alternatively you should use commercial names for gadolinium based agents as well.

ANSWER: Done. As requested, we updated the manuscript adding molecules name rather than commercial names of contrast media.

Point 2:  in the methods please mention the acquisition plane for IFIR and MRA sequences and whether any cardiac or pulse gating was necessary.

ANSWER: Done, we added in the manuscript

Point 3: in the discussion section, in order to give a comprehensive overview of this application, please mention the possibilities of IFIR sequences in other anatomic districts with appropriate references.

ANSWER: Thank you for the comment. As requested, we added a few references for other anatomic districts where IFIR can be useful and updated reference lists, accordingly.

Reviewer 2 Report

This manuscript explores the potentials for contrast agent free
magnetic resonance angiography (MRA) based on Inflow Inversion
Recovery (IFIR). The method is compared to computed tomography
angiography and MRA using contrast agents. The study involves 70
patients and concludes that IFIR-MRA is the superior method in
particular for cases where contrast agents are contraindicated.

The subject of this manuscript is at the fringe of my expertise, so I
can not comment on the novelty of the results or the soundness of the
statistical analysis. However, results would be in line with my
expectations and would extend the MR toolkit for contrast free
diagnostic methods while avoiding ionising radiation. I therefore
suggest publication after addressing some minor concerns below.

- LHA and RHA are not introduced and not in list of abbreviations.

- "DSA has certainly a small but not negligible risk of complications
  and is expensive" This requires more discussion and a reference.

- Abbreviations should be used once they are introduced: steady-state
  free-precession acquisition => SSFP at line 112/113

  Please check the manuscript for similar occurrences. 

- The abstract states that "IFIR-MRA demonstrated a higher image
  quality in main upper abdomen arterial vessels assessment". However,
  in section 3.3 it is stated that for MRA "all vessels were correctly
  evaluated in all patients" while for IFIR-MRA "common hepatic,
  gastroduodenal, left gastric and splenic arteries were correctly
  evaluated in all patients, while the celiac trunk, proper hepatic,
  and right hepatic arteries in the majority (n=69, 98.5%)." This
  would indicate that IFIR-MRA is inferior to MRA in contradiction to
  what the abstract says. What am I missing here?

  I realised that the "left and right hepatic arteries obtained the
  lowest median value 1 (IQR 1-2)" for MRA, so I think the issue is
  just the way how the statements are worded.

- This sentence is broken or incomplete: "Even if the inter-test
  reliability between IFIR-MRA and CTA was good only for the common
  hepatic artery (ICC=0.552).

Author Response

Point 1: LHA and RHA are not introduced and not in list of abbreviations.

ANSWER: In the abstract and manuscript we refer to this abbreviation “common-proper-left-right hepatic artery (C-P-L-R-HA)” for CHA, PHA, LHA, and RHA. If considered confusing, we can make it clearer with  distinct abbreviations.

Point 2:"DSA has certainly a small but not negligible risk of complications and is expensive" This requires more discussion and a reference.

ANSWER: Thank you for the comment. We accordingly updated the manuscript with the following sentence: “DSA is burdened with a greater dose of radiation and contrast media, higher cost, and invasiveness in comparison with CTA, and cannot be used for screening or routine follow-up”. We also added an appropriate reference and updated the reference lists, accordingly.

Point 3. Abbreviations should be used once they are introduced: steady-state free-precession acquisition => SSFP at line 112/113. Please check the manuscript for similar occurrences. 

ANSWER: We updated the manuscript accordingly.

Point 4: The abstract states that "IFIR-MRA demonstrated a higher image quality in main upper abdomen arterial vessels assessment". However, in section 3.3 it is stated that for MRA "all vessels were correctly evaluated in all patients" while for IFIR-MRA "common hepatic, gastroduodenal, left gastric and splenic arteries were correctly evaluated in all patients, while the celiac trunk, proper hepatic, and right hepatic arteries in the majority (n=69, 98.5%)." This would indicate that IFIR-MRA is inferior to MRA in contradiction to what the abstract says. What am I missing here?

I realised that the "left and right hepatic arteries obtained the lowest median value 1 (IQR 1-2)" for MRA, so I think the issue is just the way how the statements are worded.

ANSWER: We stated in the abstract that “IFIR-MRA demonstrated a higher image quality in main upper abdomen arterial vessels assessment”. If comparing the image quality scores among IFIR and MRA, in the first one (when evaluable) the obtained median values are always higher. However, in order to make the sentence clearer, we updated sentence in the abstract and discussion sessions.

Point 5: This sentence is broken or incomplete: "Even if the inter-test reliability between IFIR-MRA and CTA was good only for the common hepatic artery (ICC=0.552).

ANSWER: We linked the sentence with the next one, in order to make the reading more fluent.